# Association of Flow Rate of Prehospital Oxygen Administration and Clinical Outcomes in Severe Traumatic Brain Injury

**DOI:** 10.3390/jcm10184097

**Published:** 2021-09-10

**Authors:** Won Pyo Hong, Ki Jeong Hong, Sang Do Shin, Kyoung Jun Song, Tae Han Kim, Jeong Ho Park, Young Sun Ro, Seung Chul Lee, Chu Hyun Kim, Joo Jeong

**Affiliations:** 1Laboratory of Emergency Medical Services, Biomedical Research Institute, Seoul National University Hospital, Seoul 03080, Korea; pyotang@gmail.com (W.P.H.); shinsangdo@gmail.com (S.D.S.); skci-va@gmail.com (K.J.S.); adoong2001@gmail.com (T.H.K.); timthe@gmail.com (J.H.P.); Ro.youngsun@gmail.com (Y.S.R.); scl0126@hanmail.net (S.C.L.); yukijeje@gmail.com (J.J.); 2119 EMS Division, The Korean National Fire Agency, Sejong City 30128, Korea; 3Department of Emergency Medicine, College of Medicine, Seoul National University, Seoul 03080, Korea; 4Department of Emergency Medicine, Seoul National University Hospital, Seoul 03080, Korea; 5Department of Emergency Medicine, Boramae Medical Center, Seoul Metropolitan Government—Seoul National University, Seoul 07061, Korea; 6Graduate School, Dongguk University, Goyang-si 10326, Korea; 7Department of Emergency medicine, Emergency Medical Center, Dongguk University, Ilsan Hospital, Goyang-si 10326, Korea; 8Department of Emergency Medicine, Inje University College of Medicine, Seoul Paik Hospital, Seoul 04551, Korea; juliannnn@hanmail.net

**Keywords:** traumatic brain injury, prehospital, oxygenation, hypoxia, hyperoxia, emergency medical services

## Abstract

The goal of this study was to investigate the association of prehospital oxygen administration flow with clinical outcome in severe traumatic brain injury (TBI) patients. This was a cross-sectional observational study using an emergency medical services-assessed severe trauma database in South Korea. The sample included adult patients with severe blunt TBI without hypoxia who were treated by EMS providers in 2013 and 2015. Main exposure was prehospital oxygen administration flow rate (no oxygen, low-flow 1~5, mid-flow 6~14, high-flow 15 L/min). Primary outcome was in-hospital mortality. A total of 1842 patients with severe TBI were included. The number of patients with no oxygen, low-flow oxygen, mid-flow oxygen, high-flow oxygen was 244, 573, 607, and 418, respectively. Mortality of each group was 34.8%, 32.3%, 39.9%, and 41.1%, respectively. Compared with the no-oxygen group, adjusted odds (95% CI) for mortality in the low-, mid-, and high-flow oxygen groups were 0.86 (0.62–1.20), 1.15 (0.83–1.60), and 1.21 (0.83–1.73), respectively. In the interaction analysis, low-flow oxygen showed lower mortality when prehospital saturation was 94–98% (adjusted odds ratio (AOR): 0.80 (0.67–0.95)) and ≥99% (AOR: 0.69 (0.53–0.91)). High-flow oxygen showed higher mortality when prehospital oxygen saturation was ≥99% (AOR: 1.33 (1.01~1.74)). Prehospital low-flow oxygen administration was associated with lower in-hospital mortality compared with the no-oxygen group. High-flow administration showed higher mortality.

## 1. Introduction

Traumatic brain injury (TBI) is a major health and socioeconomic problem throughout the world [1]. About 5.48 million people are estimated to suffer from severe TBI each year (73 cases per 100,000 people), and the economic and social impact of TBI is considerable due to the direct and indirect costs of treatment, rehabilitation, and permanent sequelae. The World Health Organization reported the TBI global incidence is rising and was predicted to surpass many diseases as a major cause of death and disability by the year 2020 [2,3].

Prehospital hypoxia less than 90% of saturation was associated with higher mortality in previous studies, and Guidelines for the Management of Severe Traumatic Brain Injury recommends that hypoxia (partial pressure of oxygen in arterial blood (PaO2) < 60 mmHg or peripheral oxygen saturation (SpO2) < 90%) should be avoided, but there is no therapeutic range of oxygen saturation [4,5,6]. The Prehospital Trauma Life Support manual suggests that oxygen delivery should be provided based on the patient’s breathing frequency, and this tends to encourage the use of a high fraction of inspired oxygen, which results in the common use of high-flow (15 L/min) oxygen administration [7].

However, recent studies, especially in intensive care unit settings, report that not only hypoxia but hyperoxia was associated with poor outcomes [8,9,10,11]. Oxidative stress with consequent impairment of endogenous antioxidant defense mechanisms plays a significant role in the secondary events leading to neuronal death [12].

Recent study of TBI management recommends an optimal PaO2 of more than 60 mmHg and less than 200 mmHg [13]. There are no guidelines on oxygen saturation level for optimal care in the prehospital setting, and possible effects of hyperoxia from high-flow oxygenation can easily be neglected. Prehospital high-flow oxygen administration is likely not associated with poor outcome because of short transportation time. It is uncertain, however, whether high-flow oxygen administration during emergency medical services (EMS) treatment is associated with poor outcomes from TBI.

The purpose of this study was to determine the association of prehospital oxygen administration flow rate on hospital mortality and neurological outcomes in severe TBI patients without hypoxia. We hypothesized that excessive oxygenation would adversely affect survival in patients with severe TBI without hypoxia.

## 2. Materials and Methods

### 2.1. Study Design

This was a cross-sectional observational study using a database from the nationwide registry of EMS-assessed severe trauma in Korea. This national severe trauma database was built from two data sources, including the EMS severe trauma registry recorded by EMS providers and hospital medical records collected by the Korea Disease Control and Prevention Agency. The study was reviewed and approved by the institutional review board of the study institution and informed consent was waived (Approval number: 1206-024-412).

### 2.2. Study Setting

The emergency medical services system in Korea is a single-tiered public service model by the government-run fire department. The service level of prehospital care is comparable with intermediate-level emergency medical technicians in the United States. Prehospital TBI protocol includes airway management and oxygen administration to the patient with hypoxia less than 94% of saturation (SpO2 < 94%) to avoid hypoxia, but there is no clear flow rate or method of oxygen administration or target saturation level. According to capacity and resources, emergency departments in Korea are divided into levels 1 to 3, and for the patients with severe trauma, and prehospital protocol recommends transferring patients with severe TBI to a level 1 or 2 emergency department for proper management. 

### 2.3. Data Source

This study used the nationwide registry of the EMS-ST database built from the EMS severe trauma registry and hospital medical records. EMS providers used a field triage scheme consisting of four decision steps (physiologic, anatomic, mechanism of injury, and special considerations) to include patients with possible severe trauma [14], and the EMS severe trauma registry includes basic ambulance operation information and detailed prehospital monitoring and treatment information. Hospital medical records were collected by Korean CDC reviewers who received 26 h in an education course that included the coding for an abbreviated injury scale (AIS). The quality management committee, which consisted of emergency physicians, epidemiologists, statistical experts, and medical record review experts, held monthly meetings for quality assurance.

### 2.4. Selection of Participants

The study population included all patients with severe TBI who were treated by EMS providers in 10 provinces between January and December 2013 and in 17 provinces (whole country) between January and December 2015. All patients with severe blunt TBI older than 15 years old were enrolled. Severe TBI was defined according to an AIS score of 3 or above for a head lesion. Patients who had cardiac arrest at the scene, unknown prehospital oxygen saturation, prehospital hypoxia less than 94% of oxygen saturation (SpO2 < 94%), and unknown prehospital blood pressure or who had unknown information on hospital outcomes were excluded.

### 2.5. Variables and Measurements

The main exposure of interest was prehospital oxygen flow by EMS providers. Patients without oxygen administration were considered as reference, and low-flow oxygen was defined as 1~5 L/min of oxygen administration, mid-flow oxygen as 6~14 L/min, and high-flow oxygen as 15 L/min, regardless of method of oxygen supply. High prehospital oxygen saturation status was defined as more than 99% of oxygen saturation (SpO2 ≥ 99%) after oxygen administration.

Collected variables were demographic factors (age, gender, place of residence, past medical history), injury-related factors (time of trauma, place of injury, mechanism of injury (blunt or not)), prehospital factors (EMS transportation time, prehospital vital sign, and prehospital treatment, including amount of oxygen administration and prehospital oxygen saturation after oxygen administration), and hospital factors (level of emergency department, Injury Severity Score), as well as patient outcome after admission if the patient was admitted, and Glasgow Outcome Scale at hospital discharge.

### 2.6. Outcome

The primary outcome of the study was in-hospital mortality, defined as death in the emergency department or during admission, resulting from the injury. The secondary outcome was morbidity of patients, which was defined as poor according to the Glasgow Outcome Scale from 3 to 5 at hospital discharge.

### 2.7. Statistical Analysis

Descriptive analyses were performed to examine the distributions of the study variables. Counts and proportions were used for categorical variables, and medians and interquartile ranges were used for continuous variables. Categorical variables were assessed with the chi-square test, and continuous variables were compared using Mann–Whitney U tests. The *p*-values were based on a two-sided significance level of 0.05.

Adjusted odds ratios (AORs) with 95% confidence intervals (CIs) for saturation status for the study outcomes were calculated using multivariable logistic regression analysis, with no oxygen administration as the reference. The model was adjusted for gender, age, and underlying comorbidity; season and weekday; mechanism, intent, and alcohol; response time interval, scene time interval, and transport time interval; patient alertness, presence of hypotension (systolic blood pressure below 90 mmHg in prehospital setting), and level of emergency department; and Injury Severity Score from 9 to 15, 16 to 24, and above 25.

To determine the effect of hyperoxia on the patient, this study developed an interaction model with an interaction term between prehospital oxygen flow and prehospital saturation status as the final multivariable logistic model for the study outcomes. All statistical analysis was performed using SAS software, version 9.4 (SAS Institute Inc., Cary, NC, USA).

## 3. Results

A total of 35,169 patients were enrolled in the EMS-ST database during 2013 and 2015. The number of severe blunt traumatic brain injuries was 7697. After excluding ineligible patients, the final study population consisted of 1842 patients (Figure 1). Of the 1842 patients, the number of patients with no oxygen, low-flow oxygen, mid-flow oxygen, and high-flow oxygen was 244 (13.2%), 573 (31.1%), 607 (32.9%) and 418 (22.7%), respectively; the in-hospital mortality rates were 34.8%, 32.3%, 39.9% and 41.1%, respectively. Basic patient demographics are shown in Table 1. Patients were older in the no-oxygen group (median age was 61 years old) compared with other groups (median age 46, 44, and 37, respectively). Patient’s residence, mechanism of injury, patient’s alertness, prehospital hypotension, prehospital advanced airway management, prehospital IV access, and prehospital transport time were associated with the flow rate of oxygen administration (Table 1 and Table 2).

In the multivariable logistic regression analysis, low-flow oxygen administration was likely to have better in-hospital outcomes (AOR 0.86 (0.62–1.20) for in-hospital mortality and AOR 0.80 (0.57–1.10) for poor neurologic outcome) (Table 3.) High-flow oxygen administration was likely to have more mortality and poor neurologic outcome compared with the no-oxygen administration group (AOR 1.21 (0.83–1.73) for mortality and 1.15 (0.81–1.64) for poor neurological outcome).

In the interaction model, using prehospital oxygenation and prehospital saturation status, the low-flow oxygen group showed low in-hospital mortality and better neurologic outcome in both saturation groups. Adjusted odds ratio (AOR) (95% CI) for mortality was 0.80 (0.67–0.95) in the 94~98% group and 0.69 (0.53–0.91) in 99~100% group. High-flow oxygen administration showed poor in hospital outcome (AOR (95% CI) was 1.33 (1.01–1.74)) in patients with high prehospital saturation (SpO2 ≥ 99%), which was statistically significant (*p* = 0.04) (Table 4).

## 4. Discussion

Prehospital administration of oxygen is widespread in our practice, but resuscitative oxygen administration frequently exceeds the physiological needs of patients with TBI and without TBI [7,15,16]. Although this is usually accepted to avoid hypoxia, toxicity of oxygen to the brain and other vital organs due to reactive oxygen species is well described [17,18,19,20], and 100% oxygen can cause cerebral vasoconstriction, reducing cerebral perfusion [21,22].

In our study, the low-flow oxygen group showed low in-hospital mortality and better neurologic outcome. AOR (95% CI) for mortality was 0.80 (0.67–0.95) in the 94~98% group and 0.69 (0.53–0.91) in the 99~100% group in the interaction model. This result implies that low-flow (1~5 L/min) oxygen administration could be helpful for the patients with severe TBI. Recent studies have reported that oxygen administration improves cerebral metabolism and decreases intracerebral pressure [23,24], and they recommend providing normo-baric hyperoxia in the treatment of patients with TBI, but other studies have reported contrary results [6], which needs further well-controlled study.

On the other hand, 186 patients (10.1%) received high-flow oxygen, even though their prehospital saturation was above 99%; mortality among them was highest (40.9%) among all groups. Brenner et al. reported that hyperoxia, which was defined as PaO2 higher than 200 mmHg, within the first 24 h of hospitalization is associated with worse short-term functional outcomes and higher mortality after TBI [13], but other studies showed no significant difference in in-hospital mortality among patients with hyperoxia (PaO2 > 300 mmHg) [25], and there was no association between maximum PaO2 in the first 24 h after admission and in-hospital mortality [26]. This study could not measure PaO2 due to the lack of a modality to measure it exactly in prehospital settings. Additionally, the hypothesis of this investigation was that a patient could have hyperoxia when oxygen saturation was above 99% after the administration of high-flow oxygen. In the interaction analysis, patients with high oxygen saturation after high-flow oxygen showed significantly poor outcomes (AOR 1.33 (95%CI: 1.01–1.74)), which implies that hyperoxia could be harmful.

Some authors reported that prehospital advanced airway techniques were related to poor outcomes in traumatic brain injury and that this was associated with prehospital hyperventilation, which was very common (60~70%); even the prehospital guideline recommends not to hyperventilate [27,28,29,30]. In this study setting, prehospital advanced airway techniques, including laryngeal mask airway and endotracheal intubation, were uncommon (1.4%) because cases with prehospital hypoxia less than 94% were excluded. When we analyzed that separately, it did not influence our result.

The primary goal of treatment for patients with TBI is to prevent secondary brain injury. This includes providing adequate oxygenation and circulation to perfuse the brain. Oxygen should be titrated not only to prevent hypoxia but also to prevent hyperoxia. Low-flow oxygen administration could be helpful to patients with severe TBI, and indiscriminate high-flow oxygen administration could be harmful to patients with severe TBI. A more specific prehospital oxygen administration guideline (therapeutic target range of oxygen saturation 94–98% and restriction of indiscriminate high-flow oxygen) should be applied. Moreover, further study, such as a randomized controlled study, should be conducted to elucidate a clear causal relationship.

### Limitations

First, this was a cross-sectional observational study using a database from the nationwide registry of EMS-assessed severe trauma in Korea. Patients with severe TBI who visited the emergency department in their own vehicle could have been omitted from this registry. Second, the definition of hypoxia used in this study was the cutoff value of SpO2 94%, measured by pulse oximetry. The definition of hypoxia differed between the studies. Third, initiation time and duration of prehospital oxygen administration was not collected. Forth, additional physiologic parameters associated with outcome in TBI patients, such as PaO2, intracranial pressure, cerebral perfusion pressure, oxygen radicals, and cerebral metabolites, were not collected.

## 5. Conclusions

Prehospital low-flow oxygen administration was associated with low in-hospital mortality compared with the no-oxygen group in patients with severe traumatic brain injury, and high-flow oxygen administration showed higher mortality and could be harmful to patients with severe blunt traumatic brain injury. The proper therapeutic window for prehospital oxygenation may reduce the mortality rate of patients with severe TBI.

## Figures and Tables

**Figure 1 jcm-10-04097-f001:**
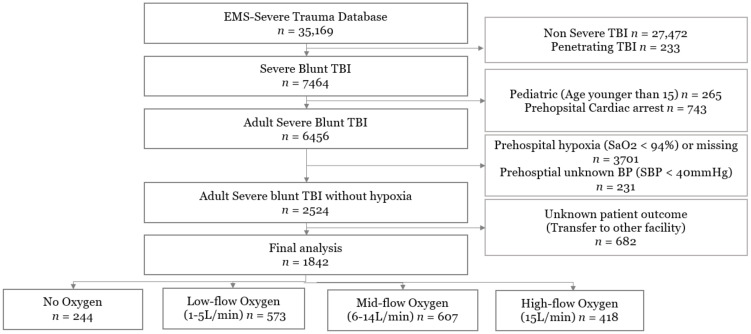
Inclusion of study population.

**Table 1 jcm-10-04097-t001:** Demographics of the study population.

		Total		No Oxygen		Flow Rate of Oxygen Administration						*p*-Value
						Low		Mid		High		
		*n*	%	*n*	%	*n*	%	*n*	%	*n*	%	
Total		1842	100	244	100	573	100	607	100	418	100	
Gender												0.16
	Male	1370	74.4	169	69.3	438	76.4	457	75.3	306	73.2	
	Female	472	25.6	75	30.7	135	23.6	150	24.7	112	26.8	
Age, years												<0.01
	15–64	1123	61.0	141	57.8	317	55.3	373	61.4	292	69.9	
	65–	719	39.0	103	42.2	256	44.7	234	38.6	126	30.1	
	Median (IQR)	58 (45–71)		61 (51–72)		46 (60–74)		44 (57–70)		37 (54–67)		
Season												0.57
	Spring	461	25.0	61	25.0	142	24.8	148	24.4	110	26.3	
	Summer	476	25.8	65	26.6	144	25.1	154	25.4	113	27.0	
	Fall	503	27.3	71	29.1	148	25.8	164	27.0	120	28.7	
	Winter	402	21.8	47	19.3	139	24.3	141	23.2	75	17.9	
Weekday												0.66
	Monday	226	12.3	35	14.3	70	12.2	76	12.5	45	10.8	
	Tuesday	228	12.4	36	14.8	73	12.7	68	11.2	51	12.2	
	Wednesday	281	15.3	36	14.8	91	15.9	91	15.0	63	15.1	
	Thursday	294	16.0	43	17.6	85	14.8	92	15.2	74	17.7	
	Friday	262	14.2	21	8.6	80	14.0	92	15.2	69	16.5	
	Saturday	255	13.8	38	15.6	79	13.8	86	14.2	52	12.4	
	Sunday	296	16.1	35	14.3	95	16.6	102	16.8	64	15.3	
Metropolis area												<0.01
	Yes	756	41.0	61	25.0	260	45.4	254	41.8	181	43.3	
	No	1086	59.0	183	75.0	313	54.6	353	58.2	237	56.7	
Mechanism												<0.01
	Traffic accident	1056	57.3	121	49.6	303	52.9	367	60.5	265	63.4	
	Fall	741	40.2	116	47.5	258	45.0	228	37.6	139	33.3	
	Other blunt	45	2.4	7	2.9	12	2.1	12	2.0	14	3.3	
Intent												0.63
	Non-intentional	1755	95.3	234	95.9	547	95.5	583	96.0	391	93.5	
	Intentional	45	2.4	6	2.5	13	2.3	13	2.1	13	3.1	
	Unknown	42	2.3	4	1.6	13	2.3	11	1.8	14	3.3	
Alcohol consumption												0.09
	Non-alcohol	54	2.9	8	3.3	18	3.1	21	3.5	7	1.7	
	Alcohol	293	15.9	41	16.8	108	18.8	91	15.0	53	12.7	
	Unknown	1495	81.2	195	79.9	447	78.0	495	81.5	358	85.6	

**Table 2 jcm-10-04097-t002:** Pre-hospital and in-hospital clinical findings according to flow rate of oxygen administration.

				Total			No Oxygen		Flow Rate of Oxygen Administration						*p*-Value
									Low		Mid		High		
				*n*	%		*n*	%	*n*	%	*n*	%	*n*	%	
Total				1842	100		244	100	573	100	607	100.0	418	100	
Patient alertness															<0.01
	Alert			284	15.4		65	26.6	122	21.3	64	10.5	33	7.9	
	Verbal			416	22.6		88	36.1	155	27.1	122	20.1	51	12.2	
	Pain			792	43.0		75	30.7	235	41.0	281	46.3	201	48.1	
	Unresponsive			350	19.0		16	6.6	61	10.6	140	23.1	133	31.8	
Prehospital SBP															<0.01
	<90 mmHg			110	6.0		6	2.5	23	4.0	41	6.8	40	9.6	
	≥90 mmHg			1732	94.0		238	97.5	550	96.0	566	93.2	378	90.4	
Prehospital Saturation															0.07
	94–98%			1054	57.2		132	54.1	354	61.8	336	55.4	232	55.5	
	99–100%			788	42.8		112	45.9	219	38.2	271	44.6	186	44.5	
Prehospital advance airway															<0.01
	No			1816	98.6		243	99.6	572	99.8	598	98.5	403	96.4	
	Yes			26	1.4		1	0.4	1	0.2	9	1.5	15	3.6	
Prehospital IV access															<0.01
	No			1520	82.5		223	91.4	488	85.2	497	81.9	312	74.6	
	Yes			322	17.5		21	8.6	85	14.8	110	18.1	106	25.4	
Response time interval (min)															0.01
	0–3			149	8.1		21	8.6	47	8.2	49	8.1	32	7.7	
	4–7			927	50.3		95	38.9	305	53.2	299	49.3	228	54.5	
	8–11			418	22.7		67	27.5	127	22.2	131	21.6	93	22.2	
	12–15			175	9.5		25	10.2	48	8.4	70	11.5	32	7.7	
	16–			173	9.4		36	14.8	46	8.0	58	9.6	33	7.9	
	Median (IQR)			7 (5–10)			8 (5–11.5)		7 (5–9)		7 (5–11)		7 (5–9)		
Scene time interval (min)															0.03
	0–3			303	16.4		32	13.1	80	14.0	114	18.8	77	18.4	
	4–7			949	51.5		113	46.3	309	53.9	317	52.2	210	50.2	
	8–11			375	20.4		58	23.8	126	22.0	106	17.5	85	20.3	
	12–15			128	6.9		27	11.1	34	5.9	37	6.1	30	7.2	
	16–			87	4.7		14	5.7	24	4.2	33	5.4	16	3.8	
	Median (IQR)			6 (4–9)			7 (4–10)		6 (4–8)		6 (4–8)		6 (4–9)		
Transport time interval (min)															<0.01
	0–3			113	6.1		3	1.2	46	8.0	40	6.6	24	5.7	
	4–7			480	26.1		33	13.5	170	29.7	148	24.4	129	30.9	
	8–11			372	20.2		42	17.2	113	19.7	131	21.6	86	20.6	
	12–15			250	13.6		31	12.7	63	11.0	98	16.1	58	13.9	
	16–			627	34.0		135	55.3	181	31.6	190	31.3	121	28.9	
	Median (IQR)			6 (4–9)			7 (4–10)		6 (4–8)		6 (4–8)		6 (4–9)		
Operation															<0.01
	No			925	50.2		148	60.7	313	54.6	272	44.8	192	45.9	
	Yes			917	49.8		96	39.3	260	45.4	335	55.2	226	54.1	
Brain Operation															<0.01
	No			1213	65.9		186	76.2	386	67.4	374	61.6	267	<0.01	
	Yes			629	34.1		58	23.8	187	32.6	233	38.4	151	36.1	
ICU admission															<0.01
	No			346	18.8		67	27.5	117	20.4	99	16.3	63	15.1	
	Yes			1496	81.2		177	72.5	456	79.6	508	83.7	355	84.9	
Ventilator apply															<0.01
	No			930	50.5		169	69.3	340	59.3	266	43.8	155	<0.01	
	Yes			912	49.5		75	30.7	233	40.7	341	56.2	263	62.9	
Co-morbidity															0.61
	No			1658	90.0		220	90.2	508	88.7	552	90.9	378	90.4	
	Yes			184	10.0		24	9.8	65	11.3	55	9.1	40	9.6	
Associated trauma other than head															<0.01
	No			1291	70.1		192	78.7	433	75.6	402	66.2	<0.01	63.2	
	Yes			551	29.9		52	21.3	140	24.4	205	33.8	154	36.8	
Injury Severity Score															<0.01
	9–15			372	20.2		81	33.2	134	23.4	105	17.3	52	12.4	
	16–25			692	37.6		95	38.9	194	33.9	220	36.2	183	43.8	
	25–			778	42.2		68	27.9	245	42.8	282	46.5	183	43.8	
Survival to discharge															0.01
	Survived			1158	62.9		159	65.2	388	67.7	365	60.1	246	58.9	
	Expired			684	37.1		85	34.8	185	32.3	242	39.9	172	41.1	
Neurologic outcome measured by Glasgow outcome scale															<0.01
	Good (1–2)			1127	61.2		152	62.3	381	66.5	356	58.6	238	<0.01	
	Poor (3–5)			715	38.8		92	37.7	192	33.5	251	41.4	180	43.1	

**Table 3 jcm-10-04097-t003:** Pre-hospital and in-hospital clinical findings according to flow rate of oxygen administration.

		Total	Outcome		Unadjusted	Adjusted Model 1 *	Adjusted Model 2 **
		*n*	*n*	%	OR (95% CI)	OR (95% CI)	OR (95% CI)
Primary outcome: in-hospital mortality							
Total		1842	684	37.1			
Oxygen flow rate (L/min)							
	No oxygen	244	85	34.8	1.00	1.00	1.00
	Low-flow rate	573	185	32.3	0.89 (0.65–1.22)	0.88 (0.64–1.21)	0.86 (0.62–1.20)
	Mid-flow rate	607	242	39.9	1.24 (0.91–1.69)	1.25 (0.92–1.70)	1.15 (0.83–1.60)
	High-flow rate	418	172	41.1	1.31 (0.94–1.82)	1.33 (0.96–1.86)	1.21 (0.83–1.73)
Secondary outcome: poor neurologic outcome							
Total		1842	715	38.8			
Oxygen flow rate (L/min)							
	No oxygen	244	92	37.7	1.00	1.00	1.00
	Low-flow rate	573	192	33.5	0.83 (0.61–1.14)	0.82 (0.60–1.13)	0.80 (0.57–1.10)
	Mid-flow rate	607	251	41.4	1.17 (0.86–1.58)	1.17 (0.86–1.59)	1.09 (0.78–1.50)
	High-flow rate	418	180	43.1	1.25 (0.90–1.73)	1.27 (0.92–1.76)	1.15 (0.81–1.64)

* Model 1: Adjusted by gender, age, underlying co-morbidity; ** Model 2: Adjusted by gender, age, underlying co-morbidity, season, weekday, mechanism, intent, alcohol, response time interval, scene time interval, transport time interval, patient alertness, low blood pressure, abnormal respiration rate, intravenous fluid, prehospital oxygen saturation status or oxygen flow.

**Table 4 jcm-10-04097-t004:** Interaction analysis for clinical outcome according to oxygen flow rate by initial prehospital oxygen saturation level.

		Total	Outcome		Adjusted OR *
		*n*	*n*	%	OR * (95% CI)
Primary outcome: in-hospital mortality					
Saturation 94–98%					
	No oxygen administration	132	48	36.4	1.00
	Low-flow rate (1–5 L/min)	354	127	35.9	0.80 (0.67–0.95)
	Mid-flow rate (6–14 L/min)	336	143	42.6	1.10 (0.94–1.29)
	High-flow rate (15 L/min)	232	96	41.4	1.18 (0.98–1.42)
Saturation 99–100%					
	No oxygen administration	112	37	33.0	1.00
	Low-flow rate (1–5 L/min)	219	58	26.5	0.69(0.53–0.91)
	Mid-flow rate (6–14 L/min)	271	99	36.5	1.05 (0.83–1.34)
	High-flow rate (15 L/min)	186	76	40.9	1.33 (1.01–1.74)
Secondary outcome: poor neurologic outcome					
Saturation 94–98%					
	No oxygen administration	132	52	40.2	1.00
	Low-flow rate (1–5 L/min)	354	132	37.3	0.78 (0.66–0.92)
	Mid-flow rate (6–14 L/min)	336	149	44.3	1.09 (0.93–1.27)
	High-flow rate (15 L/min)	232	103	44.4	1.17 (0.97–1.41)
Saturation 99–100%					
	No oxygen administration	112	39	34.8	1.00
	Low-flow rate (1–5 L/min)	219	60	27.4	0.69 (0.53–0.91)
	Mid-flow rate (6–14 L/min)	271	102	37.6	1.05 (0.83–1.34)
	High-flow rate (15 L/min)	186	77	41.4	1.29 (0.98–1.69)

* Adjusted by gender, age, underlying co-morbidity, season, weekday, mechanism, intent, alcohol, response time interval, scene time interval, transport time interval, patient alertness, low blood pressure, abnormal respiration rate, intravenous fluid, prehospital oxygen saturation status or oxygen flow.

## Data Availability

Data was obtained from the Korea Disease Control and Prevention Agency and is available from the Korea Disease Control and Prevention Agency with the permission.

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
