# Peer review of "Association of Flow Rate of Prehospital Oxygen Administration and Clinical Outcomes in Severe Traumatic Brain Injury"

_jcm, 2021, doi:10.3390/jcm10184097_

Round 1

Reviewer 1 Report

Manuscript entitled " Association of flow rate of prehospital oxygen administration and clinical outcomes in severe traumatic brain injury" is scientifically sound and well written article. The authors provide sufficient background and detailed methodology for the study. The findings are reported and discussed well. There are some minor issues that would make the manuscript easy for the readers to follow:

1) Please state a hypothesis for the study. 

2) For tables, please use bold or * to highlight the row of importance and that show statistical significance. 

3) Please expand on the implications of current findings with some discussion about what future studies would warrant. 

Author Response

Thank you for your valuable comments.

We revised our manuscript according to your suggestion.

1) Please state a hypothesis for the study. 

A) Line 131-132: “We hypothesized that excessive oxygenation would adversely affect survival in patients with severe TBI without hypoxia.”

2) For tables, please use bold or * to highlight the row of importance and that show statistical significance. 

A) Table 1, 2, 4 changed bold in case of p<0.05 or statistical significant OR

3) Please expand on the implications of current findings with some discussion about what future studies would warrant. 

A) Line 347-351: “Further delicate prehospital oxygen administration guideline(therapeutic target range of oxygen saturation 94-98% and restriction of indiscriminate high-flow oxygen) should be applied. And further study such as randomized controlled study should be conducted to elucidate a clear causal relationship.”

Reviewer 2 Report

The supply of Oxygen in the initial moment of care for patients with severe brain trauma, is a topic of high interest in Critical care and Emergencies, at the present time.

Material and Methods are adequate.

The Statistical study, is rigorous.

The Results are adequately exposed.

The Bibliographic Discussion is up-to-date.

Author Response

Thank you for your kind review.